# A high-conductivity *n*-type polymeric ink for printed electronics

Chi-Yuan Yang [1], Marc-Antoine Stoeckel [1], Tero-Petri Ruoko [1], Han-Yan Wu[1], Xianjie Liu[1], Nagesh B. Kolhe[2], Ziang Wu[3], Yuttapoom Puttisong [4], Chiara Musumeci[1], Matteo Massetti[1], Hengda Sun [1], Kai Xu[1], Deyu Tu [1], Weimin M. Chen [4], Han Young Woo [3], Mats Fahlman [1], Samson A. Jenekhe [2], Magnus Berggren [1,5,6] & Simone Fabiano [1,5,6✉]

Conducting polymers, such as the *p*-doped poly(3,4-ethylenedioxythiophene):poly(styrene sulfonate) (PEDOT:PSS), have enabled the development of an array of opto- and bio-electronics devices. However, to make these technologies truly pervasive, stable and easily processable, *n*-doped conducting polymers are also needed. Despite major efforts, no *n*-type equivalents to the benchmark PEDOT:PSS exist to date. Here, we report on the development of poly(benzimidazobenzophenanthroline):poly(ethyleneimine) (BBL:PEI) as an ethanol-based *n*-type conductive ink. BBL:PEI thin films yield an *n*-type electrical conductivity reaching 8 S cm$^{-1}$, along with excellent thermal, ambient, and solvent stability. This printable *n*-type mixed ion-electron conductor has several technological implications for realizing high-performance organic electronic devices, as demonstrated for organic thermoelectric generators with record high power output and *n*-type organic electrochemical transistors with a unique depletion mode of operation. BBL:PEI inks hold promise for the development of next-generation bioelectronics and wearable devices, in particular targeting novel functionality, efficiency, and power performance.

[1] Laboratory of Organic Electronics, Department of Science and Technology, Linköping University, Norrköping, Sweden. [2] Department of Chemical Engineering and Department of Chemistry, University of Washington, Seattle, WA, USA. [3] Department of Chemistry, College of Science, Korea University, Seoul 136-713, Republic of Korea. [4] Department of Physics, Chemistry and Biology, Linköping University, Linköping, Sweden. [5] Wallenberg Wood Science Center, Linköping University, Norrköping, Sweden. [6] n-Ink AB, 58330 Linköping, Sweden. ✉email: simone.fabiano@liu.se

Conducting polymers are an enabling technology for numerous opto- and bioelectronics applications[1]. The versatile chemical synthesis, low-cost solution processability, and exclusive mechanical robustness endow conducting polymers with broad appeal in industries such as renewable energies, sensing, and healthcare[2–4]. A prototypical conducting polymer used in many opto- and bioelectronic devices is the poly(3,4-ethylenedioxythiophene) doped with poly(styrene sulfonate), PEDOT:PSS (Fig. 1a)[5]. With >100 tons produced every year, PEDOT:PSS is the most successful hole-transporting (p-type) conducting polymer. This mixed ion-electron conductor owes its success to a high electrical conductivity, spanning over several orders of magnitude and reaching values >1000 S cm$^{-1}$, an excellent ambient stability, and commercial availability as aqueous dispersions for processing via traditional coating and printing techniques[6]. Today, PEDOT:PSS has been employed as the charge extraction/injection layer in organic solar cells[7,8] and light-emitting diodes[9,10], as well as the active material in electrochromic displays[11,12], actuators[13,14], electrochemical transistors[15–17], sensors[18,19], supercapacitors[20,21], stretchable electronics[22,23], thermoelectrics[24–26], and brain-inspired memories[27,28]. However, many opto- and bioelectronic devices and systems rely on the complementarity of both high-performance hole-transporting and electron-transporting (n-type) materials.

Although several n-type conducting polymers can be doped to high conductivity (>10 S cm$^{-1}$) after deposition, their applicability is severely restricted by the use of harmful halogenated solvents, lack of thermal, ambient, and solvent stability, as well as reliable solution processability, which often result in poor device performance (see Supplementary Table 1 for a literature survey of n-doped polymers). Various design and n-doping strategies, including planarization and stiffening of the polymer backbone[29,30], engineering of the donor–acceptor character[31,32], control of the molecular dopant counterion-polymer side-chain miscibility[33,34], and use of all-polymer blends based on ground-state electron transfer[35] are being explored. Despite great progress, n-doped conducting polymers do not yet meet the performance comparable to the best p-doped polymers[36], so that no n-type equivalents to PEDOT:PSS currently exist.

Here we report an alcohol-based n-type conductive ink for printed electronics. The n-type ink is composed of the conjugated polymer poly(benzimidazobenzophenanthroline) (BBL)[37] doped with poly(ethyleneimine) (PEI)[38], an amine-based insulating polymer (Fig. 1b). The BBL:PEI polymer mixture is prepared via the formulation of an ethanol-based ink that is processable in air through simple spray-coating. After thermal activation, the n-type BBL:PEI thin films show an electrical conductivity as high as 8 S cm$^{-1}$, as well as excellent thermal and ambient stability. We also find that the high conductivity can be retained even after washing the thin films with common organic solvents, which is particularly important for the development of multi-layered optoelectronic devices. We demonstrate the useability of this material as a printed active layer in thermoelectric generators (TEGs) with record-high power output (56 nW per p–n pair at $\Delta T = 50$ K). We further implement BBL:PEI as a mixed ion-electron conductor in organic electrochemical transistors (OECTs) and demonstrate n-type depletion mode of operation as well as new power-efficient logic devices when coupled to PEDOT:PSS-based OECTs. We anticipate that BBL:PEI inks will

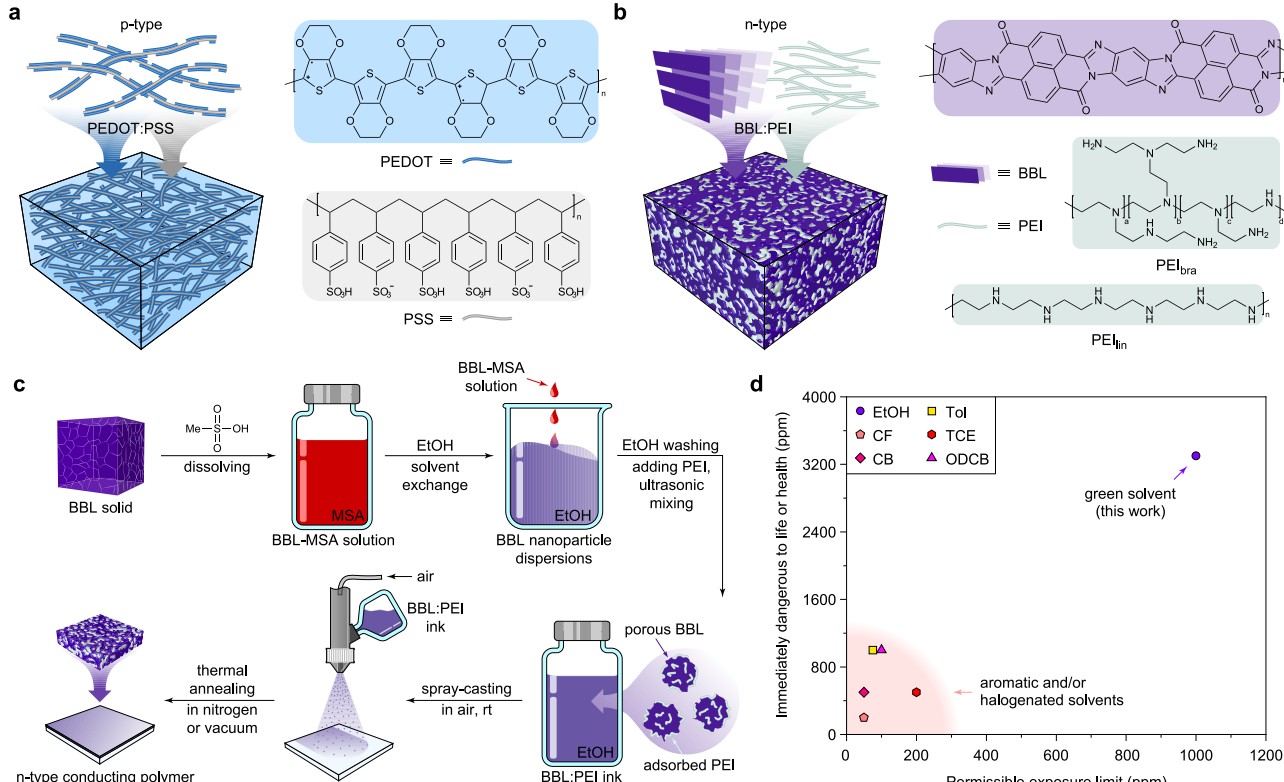

**Fig. 1 Material processing. a** Schematics of the film structure and chemical structure of p-type PEDOT:PSS. **b** Schematics of the film structure and chemical structures of BBL, PEI$_{lin}$, and PEI$_{bra}$ polymers. **c** Processing steps to obtain the BBL:PEI ink and its deposition through spray-coating. BBL powder is dissolved in MSA, followed by dispersion in ethanol through a solvent-exchange method, which results in the formation of BBL nanoparticles, and finally mixing with PEI to form the final BBL:PEI ink; BBL:PEI ink can be spray-coated in air, followed by thermal annealing, to form the highly conducting n-type film. **d** Health exposure limits of the solvent used for the BBL:PEI ink and comparison with other solvents typically used to process n-type conducting polymers (Tol, toluene; CF, chloroform; TCE, trichloroethylene; CB, chlorobenzene; and ODCB, 1,2-dichlorobenzene). Data from the US National Institute for Occupational Safety and Health (NIOSH, see Supplementary Table 2).

have a similar impact on the field of organic electronics as the prototypical PEDOT:PSS.

## Results

**Ink formulation and processing.** BBL:PEI ink was prepared by dissolving BBL in methanesulfonic acid (MSA), followed by solvent-exchange with ethanol under rapid stirring (Fig. 1c). This yields BBL nanoparticles with a size of about 20 nm in diameter (Supplementary Fig. 1). The BBL nanoparticles are then washed in ethanol and mixed with either linear PEI (PEI$_{lin}$) or branched PEI (PEI$_{bra}$) at different mass ratios, and sonicated to obtain the ethanol-based all-polymer conductive ink (see "Methods" for further details). The resulting ethanol-based ink is composed of BBL:PEI nanoparticles with a size of 30–100 nm, depending on the PEI content (Supplementary Fig. 1). The ink can be processed in air and deposited over a large area by simple spray-coating technique. Because of the low boiling point of ethanol, BBL:PEI does not require thermal annealing for drying the films, whereas a thermal treatment under inert atmosphere or vacuum is needed to reach the high-conductivity values (vide infra). In addition, unlike the aromatic and halogenated solvents typically used to process $n$-doped conducting polymers, the use of green solvent such as ethanol is expected to facilitate the transition of BBL:PEI ink from lab to fab (Fig. 1d and Supplementary Table 2).

**Film microstructure characterization.** We carried out grazing-incidence wide-angle X-ray scattering (GIWAXS) to determine the microstructure of the spray-coated BBL:PEI$_{lin}$ films (Fig. 2a–e and see Supplementary Figs. 2 and 3 for BBL:PEI$_{bra}$ films). Pure BBL is primarily oriented edge-on on the substrate, exhibiting a strong lamellar (100) peak at $q_z = 0.794$ Å$^{-1}$ ($d$-spacing = 7.91 Å) and a strong $\pi$–$\pi$ stacking (010) peak at $q_{xy} = 1.846$ Å$^{-1}$ ($d$-spacing = 3.40 Å). The diffraction pattern of pure PEI$_{lin}$ shows very sharp rings that disappear when PEI$_{lin}$ is blended with BBL, indicative of a good intermixing between the two polymeric phases. As the PEI$_{lin}$ content

increases, the BBL $\pi$–$\pi$ stacking distance decreases, reaching a value of 3.36 Å, which is indicative of a stronger $\pi$–$\pi$ interaction. This leads to higher $\pi$–$\pi$ stacking crystallinity with longer coherence lengths and lower paracrystalline disorder (Supplementary Figs. 4–6). At 50 wt% PEI$_{lin}$ content, two sets of lamellar diffraction peaks arise: (i) (100) diffraction peak with a packing distance of 7.23 Å, which is slightly smaller than that of pristine BBL, and (ii) (100)' diffraction peak with a packing distance of 11.7 Å, which is 3.79 Å longer than that of pure BBL (Supplementary Figs. 7–9). This suggests that PEI chains intercalate between the BBL lamella without directly interfering with the BBL chain $\pi$–$\pi$ stacking. The addition of PEI results in a smoother film surface, as compared to pure BBL films (Fig. 2g, i). Conductive atomic force microscopy reveals the presence of conductive regions in BBL:PEI$_{lin}$ films, as opposed to the non-conductive surface of pure BBL films (Fig. 2h, j and see Supplementary Fig. 10 for BBL:PEI$_{bra}$ films). X-ray photoelectron spectroscopy (XPS) reveals the presence of both negatively charged BBL and positively charged PEI chains (Fig. 2f and see also Supplementary Fig. 11 for a detailed N(1$s$) spectra analysis).

**Electrical performance.** The electrical conductivity of BBL:PEI thin films is reported in Fig. 3a as a function of PEI content. Pure BBL films have an electrical conductivity as low as $10^{-5}$ S cm$^{-1}$. When blended with PEI$_{lin}$ and annealed, the electrical conductivity increases to $0.10 \pm 0.02$ S cm$^{-1}$ at 5 wt% PEI and saturates to $7.7 \pm 0.5$ S cm$^{-1}$ for 50 wt% PEI content. Because of the high density of electron-donating secondary amine groups in PEI[39,40], a 5 min treatment at 150 °C is enough to reach 1 S cm$^{-1}$ $n$-type conductivity, whereas the maximum conductivity of 8 S cm$^{-1}$ is obtained after 2 h of annealing (Supplementary Fig. 12). For comparison, we measured BBL films blended with PEI$_{bra}$, bearing primary, secondary, and tertiary amines, and obtained a maximum conductivity of $4.0 \pm 0.1$ S cm$^{-1}$ for 50 wt% PEI content (Fig. 3a). When BBL is blended with polyethylenimine ethoxylated (PEIE), it reaches a conductivity of $1.4 \pm 0.1$ S cm$^{-1}$

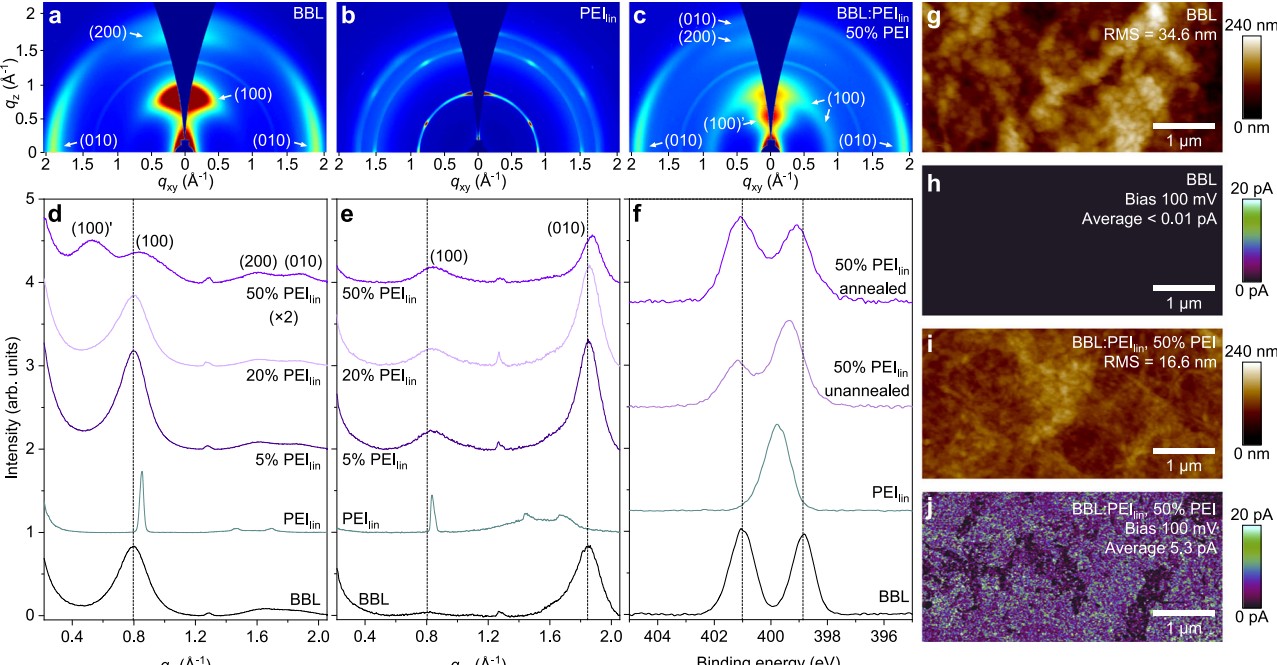

**Fig. 2 Material characterization. a–c** 2D GIWAXS patterns of BBL (**a**), PEI$_{lin}$ (**b**), and BBL:PEI$_{lin}$ (50 wt% PEI, **c**) films. **d, e** Out-of-plane (**d**) and in-plane (**e**) GIWAXS line cuts of BBL, PEI$_{lin}$, and BBL:PEI$_{lin}$ films. **f** N(1$s$) XPS analysis of BBL, PEI$_{lin}$, and BBL:PEI$_{lin}$ (50 wt% PEI) films. **g–j** Atomic force microscope (AFM) images and corresponding conductive-AFM images of BBL (**g** height image; **h** current image) and BBL:PEI$_{lin}$ (**i** height image; **j** current image). The root mean square (RMS) surface roughness and average current are also reported.

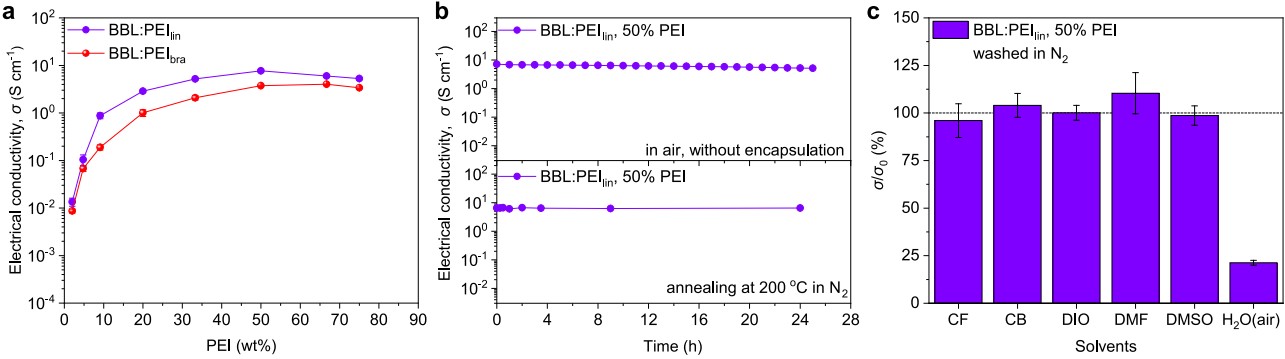

**Fig. 3 Electrical properties and stability. a** Conductivity of BBL:PEI$_{lin}$ and BBL:PEI$_{bra}$ for different PEI content. **b** Evolution of conductivity of BBL:PEI$_{lin}$ in air for 24 h and thermal stability when subjected to annealing at 200 °C in nitrogen atmosphere over 24 h. **c** Stability of BBL:PEI$_{lin}$ exposed to common organic solvents (chloroform (CF), chlorobenzene (CB), 1,8-diiodooctane (DIO), dimethylformamide (DMF), and dimethylsulfoxide (DMSO)), showing changes in conductivity. Error bars indicate the SD of ten experimental replicates.

at BBL:PEIE 5:1, with higher PEIE content leading to a degradation of the electrical performance (Supplementary Fig. 13). Prior to thermal annealing, BBL:PEI films are stable for at least 2 days in air and do not suffer from any conductivity degradation (Supplementary Fig. 14). The conductivity is independent of film thickness ($d$) for $d > 50$ nm (average ~6 S cm$^{-1}$), whereas it decreases to ~1.5 S cm$^{-1}$ when $d$ approaches the BBL nanoparticle size (i.e., for 20 nm-thick samples, Supplementary Fig. 15). The out-of-plane electrical conductivity of BBL:PEI films is 0.1 S cm$^{-1}$, which is less than two orders of magnitude lower than the in-plane conductivity (Supplementary Fig. 15). We interpret this anisotropic conductivity in terms of a percolating cluster model, developed for similar two-phase systems such as PEDOT:PSS[41]. The negatively charged BBL chains are preferentially ordered parallel to the substrate and are compensated by the long positively charged PEI chains, so that the latter are expected to have a preferential order parallel to the substrate as well. This anisotropy favors the in-plane conductivity as also observed in the case of PEDOT:PSS[41]. It is noteworthy that BBL:PEI forms an ohmic contact to various electrodes with work functions ranging from 5.1 eV (PEDOT:PSS) to 2.8 eV (Ca/Al) (Supplementary Fig. 16).

BBL:PEI shows excellent ambient stability, with the conductivity of 12 μm-thick films decreasing only about 20% upon 24 h exposure to air (Fig. 3b). This stability is induced by the combination of high work function (4.19 eV, vide infra) and self-encapsulation in the micrometer-thick film, which inhibits the penetration of H$_2$O and O$_2$, as also observed for other recently reported $n$-doped polymers[42]. The electrical conductivity of thinner films (~100 nm) drops to 0.1 S cm$^{-1}$ after 10 days in air, whereas we observed a decrease of <10% over 120 days when the thin films are stored in inert atmosphere (Supplementary Fig. 17). We also tested the thermal stability of BBL:PEI, which is crucial for applications that require continuous operation at high temperatures, such as solar cells or thermoelectrics. Remarkably, we did not observe any degradation of the conductivity (or Seebeck coefficient) even after annealing for 24 h at 200 °C in inert atmosphere (see Fig. 3b and Supplementary Fig. 18). Also, cycling the temperature between 20 °C and 100 °C did not induce any signs of degradation even after 10 cycles, whereas high-temperature annealing at 350 °C decreased the conductivity of BBL:PEI$_{lin}$ by only 30% after 80 min (Supplementary Fig. 17).

We further tested the ability of BBL:PEI to maintain the high-conductivity performance after washing the thin films with common organic solvents (Fig. 3c and Supplementary Table 3). This is particularly important for the development of multi-layer films that can be used in optoelectronic devices. BBL:PEI films can be washed

with chloroform (CF), chlorobenzene (CB), 1,8-diiodooctane (DIO), dimethylformamide (DMF), and dimethylsulfoxide (DMSO) without affecting the conductivity, which is in striking contrast to other high-performance conducting polymers doped with small molecular dopants (Supplementary Fig. 19). We chose to test these solvents, as they are routinely used to process organic and hybrid active layers in solar cells and light-emitting diodes[43,44]. A more pronounced degradation of the conductivity is observed when BBL:PEI films are washed with water, with the conductivity decreasing by less than one order of magnitude to 1.5 ± 0.1 S cm$^{-1}$. These conductivity values are in line with those of a PEDOT:PSS formulation typically used in solar cells (0.1–1 S cm$^{-1}$)[45]. The ambient, thermal, and solvent stability of BBL with PEI$_{bra}$ is reported in Supplementary Fig. 20.

**Spectroscopic characterization.** To gain insight into the nature of charge carriers that contribute to transport in BBL:PEI, we performed ultraviolet photoelectron spectroscopy (UPS) and electron paramagnetic resonance (EPR) spectroscopy on BBL:PEI films with different PEI content (Fig. 4a and Supplementary Fig. 21). The work function of pure BBL is 4.32 ± 0.05 eV and shifts to 4.19 ± 0.05 eV when the PEI content increases to 50 wt%. The EPR spectra corroborate the presence of polaronic species upon blending with PEI (Fig. 4a). The EPR spectra are reported in Supplementary Fig. 22 and the extracted data are presented in Supplementary Table 4. Although no EPR signal is detected for pure BBL, a strong EPR signal intensity is observed for BBL:PEI, reaching a spin density of $8 \times 10^{19}$ cm$^{-3}$ for BBL:PEI with 50 wt% PEI content.

The ultraviolet-visible (UV-Vis) difference absorption spectra of BBL:PEI$_{lin}$ at various PEI content are reported in Fig. 4b. These spectra were obtained by subtracting the normalized absorption spectrum of BBL from the normalized BBL:PEI spectra presented in Supplementary Fig. 23, as PEI does not absorb in the reported wavelength range. Three polaronic absorption bands grow with increasing PEI content, with a sharp band at 400 nm and two overlapping wider bands centered around 725 and 890 nm. This is consistent with previous $n$-doping[30] and spectroelectrochemical studies[46,47] of BBL, corroborating the reduction of BBL chains in contact with PEI. The annealing of BBL:PEI increases the polaronic absorption drastically, showing that the polaron formation is thermally activated, although a weaker polaronic absorption is visible also for the unannealed blends (Supplementary Fig. 24). Polaron formation in BBL is also observed in Fourier-transformed infrared (FTIR) absorption spectra recorded on the same films, in which a broad polaronic absorption band,

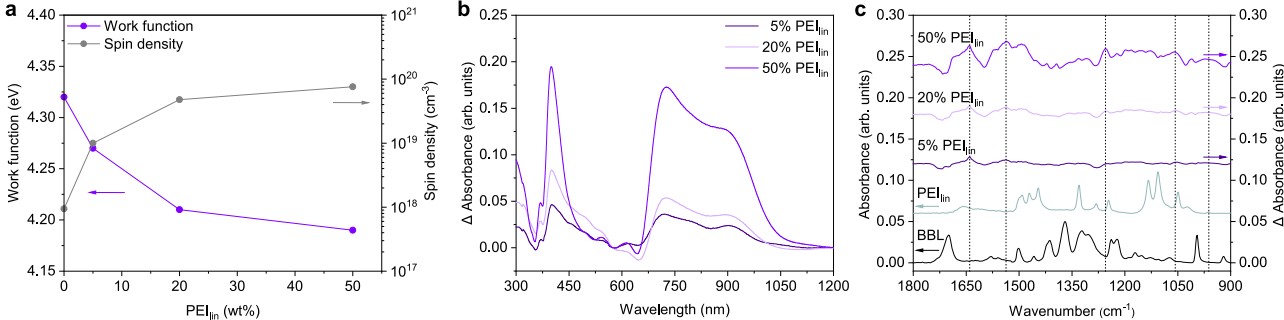

**Fig. 4 Spectroscopic confirmation of doping. a** Evolution of the work function measured by UPS and the spin density measured by EPR for BBL:PEI$_{lin}$ films with different PEI content. **b** Difference UV-Vis absorption spectra of BBL:PEI$_{lin}$ (50 wt%) showing the evolution of polaron absorption induced by the increased PEI content. **c** The corresponding difference FTIR absorption spectra of the same samples. The difference absorption spectra in **b** and **c** were obtained by subtracting the BBL spectrum from the BBL:PEI$_{lin}$ spectra after normalization at 575 nm and 1370 cm$^{-1}$ for UV-Vis and FTIR, respectively. A similar analysis is reported in Supplementary Fig. 26 for BBL:PEI$_{bra}$.

with a maximum between 3500 and 3000 cm$^{-1}$, appears upon increasing the PEI content (Supplementary Fig. 25). A similar absorption band has been previously reported for electrochemically doped BBL and is ascribed to an electronic transition in reduced BBL[46]. The difference FTIR absorption spectra in Fig. 4c highlight the emergence of five vibrational absorption bands induced by PEI that do not overlap with an existing vibration in either polymer; two strong polaronic peaks located at 1640 and 1535 cm$^{-1}$, and three weaker peaks at 1255, 1055, and 970 cm$^{-1}$. The band at 1640 cm$^{-1}$, which is the fingerprint of negative polarons in BBL[48], is assigned to antisymmetric C=O stretching and coincides with a decrease in the intensity of the C=O vibration in pristine BBL at 1700 cm$^{-1}$. On the other hand, we assign the new polaronic band at 1535 cm$^{-1}$ to C=C vibrations in BBL. Both the decrease of the 1700 cm$^{-1}$ and the increase of the 1535 cm$^{-1}$ bands are consistent with the formation of resonance-stabilized quinone or hydroquinone structures due to the reduction of BBL[47]. FTIR measurements of unannealed and annealed BBL:PEI also prove that the polaron formation is mainly thermally activated, with the C=O vibration of pristine BBL disappearing completely for annealed BBL:PEI (Supplementary Fig. 24). The three weaker vibrational peaks may further be associated with changes in BBL or reflect changes occurring in the C–N vibration in PEI due to a positive charge residing in the amine. A vibrational change in PEI is most supported by the formation of the 1255 cm$^{-1}$ peak, which lies close to the C–N stretching region of both aliphatic and aromatic amines. However, sharp polaronic absorption bands at 1255 and 1066 cm$^{-1}$ have been previously reported for electrochemically reduced BBL as well[46], indicating that assigning the weaker newly formed peaks here conclusively to changes in either BBL or PEI is not reasonable due to the overlap in the vibrational energies between the two polymers.

**Organic TEGs**. We then tested the thermoelectric properties of BBL:PEI films. The Seebeck coefficient decreases from −482 ± 1 to −65 ± 1 μV K$^{-1}$ with increasing PEI content (Fig. 5a and Supplementary Figs. 27 and 28). The negative sign of the Seebeck coefficient confirms the blend's n-type character. BBL:PEI presents a maximum power factor larger than 11 μW m$^{-1}$ K$^{-2}$ with 33 wt% PEI$_{lin}$ content (Supplementary Table 1). We then demonstrated a flexible all-polymer TEG based on BBL:PEI (33 wt% PEI$_{lin}$) n-leg and PEDOT:PSS p-leg. The resulting TEG yields short-circuit currents and open-circuit voltages that are linearly proportional to the temperature gradient, with the TEG having an inner resistance of around 200 Ω and a Seebeck

coefficient of 131 μV K$^{-1}$ (Supplementary Fig. 29). By connecting various load resistances (Fig. 5b and Supplementary Fig. 30), we measured a power output ranging from 0.54 nW ($\Delta T = 5$ K) to 56 nW ($\Delta T = 50$ K), which follows a square relationship with the temperature gradient. The power output per thermocouple is the highest reported value for in-plane all-polymer TEGs and much higher than organic TEGs with only a p-leg and metal connection (Fig. 5c and Supplementary Table 5). We also obtained similar results by printing silver electrodes (Supplementary Figs. 31 and 32), which enables the manufacturing of all-printed TEGs.

**Organic electrochemical transistors**. Finally, we tested BBL:PEI as an n-type organic mixed ionic-electronic conductor in OECTs. As BBL:PEI films are conductive in their pristine state, the resulting n-type OECT operates in the depletion mode. Figure 5d shows the typical transfer characteristics of a BBL:PEI-based OECT (50 wt% PEI$_{lin}$, transfer curve cycling and output curves are shown in Supplementary Fig. 33). It is noteworthy that in analogy with PEDOT:PSS-based OECTs, we used BBL:PEI as both the channel and gate material. The source-drain current is high at zero gate voltage and decreases by three orders of magnitude when a negative voltage bias is applied to the gate. The maximum transconductance is 0.38 mS at zero gate voltage. In addition, the device shows excellent cycling stability and fast response times ($\tau_{on} = 167$ ms and $\tau_{off} = 11$ ms, Supplementary Fig. 34). It is worth noting that n-type depletion-mode OECTs have not been realized before (see Fig. 5e and Supplementary Table 6 for a survey of the field) and their demonstration complements the current OECT technologies by bringing a new paradigm of logic circuitry. As an example, we paired an n-type BBL:PEI-based OECT to a p-type PEDOT:PSS-based OECT, both working in the depletion mode of operation, and demonstrated OECT-based ternary logic gates. This balanced ternary inverter can process three bits of information ("+1," "0," and "−1"). The electrical characteristics of the p-type PEDOT:PSS-based OECTs are reported in Supplementary Fig. 33. Figure 5f shows the ternary inverter layout and its voltage transfer characteristics, with a clear balanced ternary logic operation (Supplementary Fig. 35). During the two transition periods, the inverter exhibits similar gain values of up to 6 at $V_{in} = -0.07$ V and 0.38 V, while consuming <10 μW. For comparison, unipolar binary inverters based on printed PEDOT:PSS OECTs consume about 1 mW[17]. The dashed line in Fig. 5f shows the simulated voltage transfer characteristics of the ternary inverter, which are in good agreement with the measured data (see "Methods" and Supplementary Fig. 36 for information regarding the SPICE model used for the simulation). It is noteworthy that our OECT-based ternary

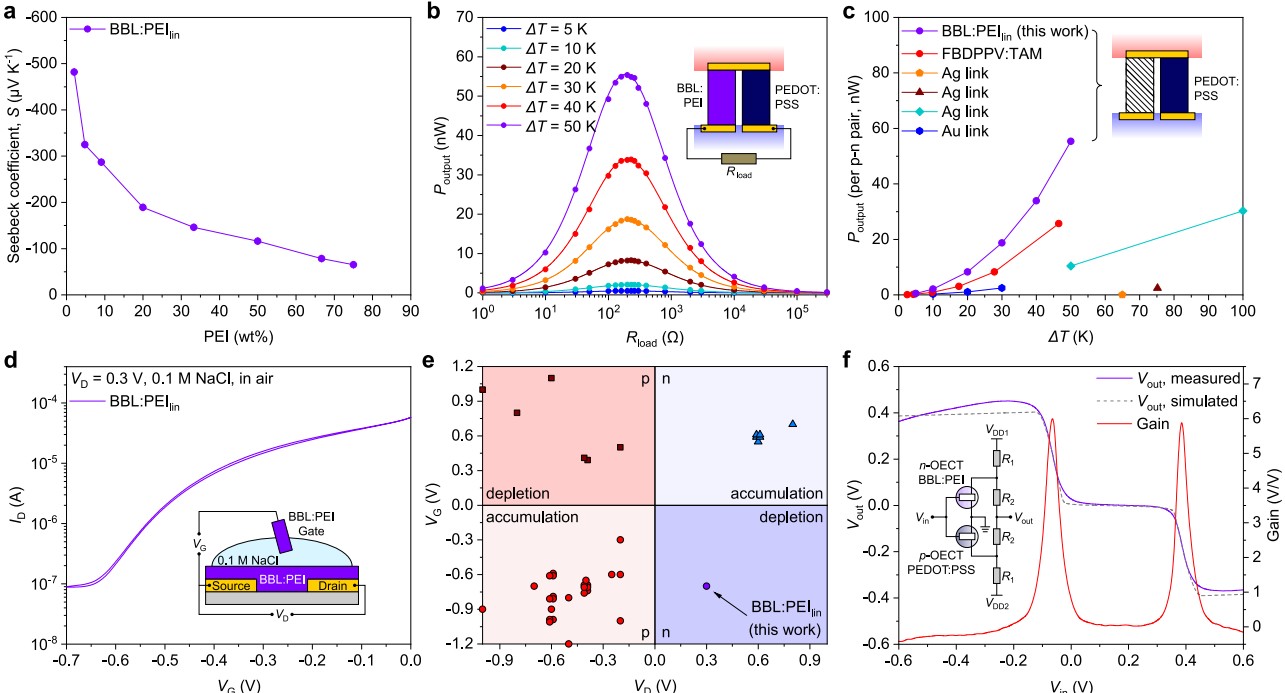

**Fig. 5 Applications. a** Seebeck coefficient of BBL:PEI$_{lin}$. **b** Power output vs. resistance loads recorded at different temperature gradients for a planar thermoelectric module composed of a PEDOT:PSS $p$-type leg and a BBL:PEI $n$-type leg. **c** Power output per $p$–$n$ pair vs. temperature gradient in this work, compared with other planar thermoelectric modules presented in literature. **d** Transfer curve of BBL:PEI-based OECT measured in air with 0.1 M NaCl electrolyte, demonstrating $n$-type conduction behavior in depletion mode. **e** Gate voltage-drain voltage map of the different regimes observed for organic semiconductor-based OECTs with both types of conduction from the literature (Supplementary Table 6). **f** Voltage output, gain, and simulated behavior of the ternary inverter integrating a PEDOT:PSS OECT for the $p$-side and BBL:PEI OECT for the $n$-side. Error bars indicate the SD of ten experimental replicates.

inverters outperform similar silicon-based tunneling ternary inverters in terms of voltage gain (Supplementary Fig. 37)[49].

## Discussion

In summary, we developed a high-conductivity ethanol-based $n$-type conductive ink with striking performance and stability, which holds great promise for printed electronics, energy technology, and bioelectronics. The negative charges on the conjugated polymer BBL are compensated by the positive charges on the amine-based insulating polymer PEI, resulting in an $n$-doped polymer–polymer ink that is processable in air through simple spray-coating. BBL:PEI thin films show an electrical conductivity as high as 8 S cm$^{-1}$, as well as excellent thermal, ambient, and solvent stability. We demonstrated the application of this material as the active layer in printed TEGs, exhibiting record high power output (56 nW per $p$–$n$ pair at $\Delta T = 50$ K) when combined with the equivalent $p$-type counterpart PEDOT:PSS. We also explored the mixed ion-electron conductor properties of BBL:PEI in OECTs and demonstrated $n$-type depletion mode of operation as well as power-efficient logic devices when coupled to PEDOT:PSS-based OECTs. We anticipate that the $n$-type BBL:PEI will have a similar impact on the field of organic electronics as the prototypical $p$-type PEDOT:PSS and will offer solutions to currently unsolved problems where complementary hole and electron transport is required, with tremendous potential impact in next-generation organic opto- and bioelectronic devices.

## Methods

**Materials**. BBL was synthesized following a procedure reported previously[50]. In brief, polyphosphoric acid (250 g) was added to a 500 mL three-necked flask fitted with an overhead stirrer and nitrogen inlet/outlet. The polyphosphoric acid was deoxygenated by heating overnight at 110 °C with nitrogen bubbling through the

stirred acid. Then, 1.92 g (6.75 mmol) of 1,2,4,5-tetraaminobenzene tetra-hydrochloride was added, under a nitrogen atmosphere at 50 °C. The mixture was heated overnight at 75 °C and 1,4,5,8-naphthalenetetracarboxylic dianhydride (1.81 g, 6.75 mmol) was then added. The mixture was slowly heated (4 °C min$^{-1}$) to 180 °C and maintained at that temperature for 10 h. The resulting viscous solution was poured out of the flask at 180 °C into a beaker and allowed to cool to room temperature. The polymer was precipitated in methanol, using a blender to facilitate mixing. The fibrous brown material was washed twice with methanol and water, and dried at 200 °C under reduced pressure (0.40 mm Hg). Precipitation from 500 g of MSA and drying in the above manner gave BBL as dark purple fibers with metallic luster (2.15 g, 95% yield, $\eta = 11.6$ dL g$^{-1}$ in MSA at 30 °C, $M_w = 60.5$ kDa). PEI$_{lin}$ ($M_n = 2.5$ kDa, polydispersity index (PDI) < 1.3), PEI$_{bra}$ ($M_n = 10$ kDa, PDI = 1.5), MSA, and ethanol were purchased from Sigma-Aldrich and were used as received. PEDOT:PSS (Clevios PH1000) was purchased from Heraeus Holding GmbH.

**Sample preparation**. The BBL:PEI ink was fabricated through a surfactant-free method. BBL (150 mg) was dissolved in MSA (75 mL) to form a deep red solution, then the BBL-MSA solution was added dropwise to ethanol (300 mL) under high-speed stirring (1500 r.p.m.). During the solvent-exchange, dark purple BBL nanoparticles were generated. The BBL nanoparticles were collected by centrifugation (1700 × g, 30 min) and washed with ethanol for six times until neutral. The neutral BBL nanoparticles were re-dispersed in ethanol to obtain a dispersion (about 1 mg mL$^{-1}$). PEI was then added to the BBL nanoparticle dispersion and the mixture was further homogenized in ultrasonic bath for 1 h to form the final BBL:PEI ink. The BBL:PEI ink can be further diluted for casting thin films in various thickness. BBL:PEI thin films were fabricated by spray-casting in air, by means of a standard HD-130 air-brush (0.3 mm) with atomization air pressure of 2 bar. After spray-casting, the BBL:PEI thin films were annealed at 140 °C for 2 h inside a nitrogen-filled glovebox or under vacuum to get the conducting film.

**UPS and XPS spectroscopy**. XPS experiment was carried out in a Scienta ESCA 200 system with a base pressure of $2 \times 10^{-10}$ mbar equipped with an SES 200 electron analyzer, a monochromatic Al Ka X-ray source (1486.6 eV), and a helium discharge lamp (21.22 eV) for XPS and UPS, respectively. All spectra were collected at normal emission and were calibrated by a sputter-cleaned Au film with the Fermi level at 0 eV and the Au(4$f$) peak at 84.0 eV. The work function was extracted from the edge of the secondary electron cutoff in UPS while applying

a −3 V bias on the sample. To extract the underlying mechanism of the charge transfer between $PEI_{lin}$ and BBL, i.e., negatively charged BBL and positively charged $PEI_{lin}$ in the mixed sample, the XPS peak of N(1s) was deconvoluted. The N(1s) spectral features obtained from fitting of pristine $PEI_{lin}$ and BBL films were used in the deconvolution of the blend samples, with additional features added corresponding to charged species created upon mixing $PEI_{lin}$ and BBL.

**Electron paramagnetic resonance.** Quantitative EPR experiments were performed at the Swedish Interdisciplinary Magnetic Resonance Centre at Linköping University, using a Bruker Elexsys E500 spectrometer operating at about 9.8 GHz (X-band). EPR spectra were recorded in dark at room temperature.

**AFM and c-AFM microscopy.** Atomic force microscopy (AFM) and conductive-AFM (c-AFM) were performed in a Dimension 3100/Nanoscope IV system, equipped with a c-AFM module (current sensitivity 1 nA V$^{-1}$) from Bruker. Soft Pt/Cr-coated silicon probes ($k = 0.2$ N m$^{-1}$) were used to simultaneously map topography and current in contact mode at a constant load force of 2–5 nN. The current maps were obtained by constantly biasing the Au substrate, while keeping the scanning AFM probe at ground. All the measurements were performed at room temperature in ambient atmosphere.

**Grazing-incidence wide-angle X-ray scattering.** GIWAXS experiments were performed at Beamline 9A at the Pohang Accelerator Laboratory in South Korea. The X-ray energy was 11.07 eV and the incidence angle was 0.12°. Samples were measured in vacuum and total exposure time was 10 s. The scattered X-rays were recorded by a charge-coupled device detector located 221.7788 mm from the sample. All samples for GIWAXS measurements had similar thickness of around 100 nm.

**UV-Vis-near infrared and FTIR.** BBL, PEI, and BBL:PEI films were prepared on calcium fluoride windows following the above-mentioned procedure for optical characterization. All measurements were performed with the film inside an air-tight sample holder, which was sealed in a nitrogen-filled glovebox. UV-Vis-near infrared absorption spectra of the films were measured with Perkin Elmer Lambda 900 with a resolution of 2 nm. The FTIR spectra were measured in transmission mode with Bruker Equinox 55, averaging 200 scans with a resolution of 4 cm$^{-1}$ and zerofilling factor of 2.

**Electrical characterization.** Electrical conductivity and Seebeck coefficient measurements were performed inside a nitrogen-filled glovebox using a Keithley 4200-SCS semiconductor characterization system. Five nanometers of chromium as an adhesive layer and 50 nm of gold were thermally evaporated on cleaned glass substrates through a shadow mask, forming electrodes with a channel length/width of 30 μm/1000 μm for the electrical and 0.5 mm/15 mm for Seebeck coefficient characterizations. Four-probe conductivity measurements were also performed and showed comparable resistances to the two-probe measurements (Supplementary Fig. 38).

**Thermoelectric generators.** The TEGs had an in-plane geometry with one p/n-leg pair module prepared on a 25 μm-thick polyethylene naphthalate (PEN) substrate. For the p-leg, we used PEDOT:PSS (PH1000) treated with DMSO (5 wt%). Considering the different electrical conductivity of secondary-doped PEDOT:PSS and BBL:PEI, the widths of the p/n legs were set to 2.5 mm/20 mm, respectively; the leg lengths and thicknesses were both 2.5 mm and 10 μm, respectively. First, the chromium/gold (5 nm/50 nm) electrodes were evaporated on the PEN substrate through a shadow mask. Then, PEDOT:PSS and BBL:PEI legs were printed through spray-coating the respective dispersions in air. The samples were then annealed in nitrogen at 140 °C and was followed by encapsulation with CYTOP. For TEGs with silver electrodes, the PEDOT:PSS and BBL:PEI legs were directly printed on PEN substrate in air and the silver paste was printed on the top of legs to form the electrodes. The samples were then annealed and encapsulated using the same method.

**OECTs and ternary inverters.** OECTs had a lateral-gate geometry. OECTs were fabricated on glass substrates (standard microscope glass). The substrates were washed by acetone, water, and isopropanol sequentially in ultrasonic bath and dried by nitrogen. Then chromium/gold (5 nm/50 nm) were deposited on the substrates through shadow mask to form the source/drain electrodes with channel length $L = 30$ μm and channel width $W = 1$ mm. For n-type depletion mode OECT, a 50 nm-thick BBL:PEI channel and gate layer were spray-coated through shadow mask with gate size of 5 mm × 5 mm. The samples were annealed in nitrogen at 140 °C for 2 h and, finally, a protection tape insulating layer were added. For the p-type depletion mode OECT, PEDOT:PSS (containing 1 wt% of (3-glycidyloxypropyl)trimethoxysilane[51] and 5 wt% of ethylene glycol) was homogenized in ultrasonic bath for 30 min and spin-coated at 4000 r.p.m. on the substrate. The PEDOT:PSS layer were patterned by protection tape to form the channel and gate (gate size of 5 mm × 5 mm). The samples were annealed at 120 °C in air for 1 min and dipped into TAM[52] ethanol solution (5–20 mg mL$^{-1}$) for 1 min. After annealing in nitrogen at 140 °C for 60 min, the samples were finally insulated by using a protection tape. For ternary inverter, one n-type

OECT, one p-type OECT, and four resistors ($R_1 = 820$ kΩ and $R_2 = 330$ kΩ, Fig. 5e) were integrated through silver paste lines. The n-type, p-type OECTs, and ternary inverters were tested in air with 0.1 M NaCl aqueous electrolyte.

**SPICE model.** SPICE models of the depletion-mode n-/p-type OECTs are developed in B2 SPICE (EMAG Technologies, Inc.). Both n-type and p-type models are built to simulate the voltage transfer characteristics of the ternary inverter. The model parameters in the sub-circuits (resistors, capacitors, and diodes) are the same for both types of OECTs, except for the polarity and threshold voltage of the depletion-mode transistors.

## Data availability
The authors declare that the main data supporting the findings of this study are available within the paper and its Supplementary Information files. Source data are provided with this paper.

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

## Acknowledgements

We thank Duyen K. Tran (U. Washington) for helpful discussion and Qilun Zhang (Linköping U.) for assistance with the dynamic light scattering measurements. This work was financially supported by the Knut and Alice Wallenberg foundation, the Swedish Research Council (2016-03979 and 2020-03243), ÅForsk (18-313 and 19-310), Olle Engkvists Stiftelse (204-0256), VINNOVA (2020-05223), and the Swedish Government Strategic Research Area in Materials Science on Functional Materials at Linköping University (Faculty Grant SFO-Mat-LiU 2009-00971). H.Y. Woo acknowledges the financial support from the National Research Foundation of Korea (NRF2020M3H4A3081814 and 2019R1A6A1A11044070). Work at the University of Washington was supported by the National Science Foundation (DMR-2003518). We acknowledge MAX IV Laboratory for time on Beamline SPECIES-APXPS under Proposal 20200356. Research conducted at MAX IV, a Swedish national user facility, is supported by the Swedish Research council under contract 2018-07152, the Swedish Governmental Agency for Innovation Systems under contract 2018-04969, and Formas under contract 2019-02496.

## Author contributions

S.F. conceived and designed the experiments. C.-Y.Y. developed the BBL:PEI ink. C.-Y.Y., M.-A.S., H.-Y. Wu, and K.X. performed the electrical characterization measurements and analyzed the data. T.-P.R. recorded and analyzed the UV-Vis-NIR and FTIR data. C.-Y.Y., N.B.K., and S.A.J. synthesized BBL. Z.W., C.-Y.Y., and H.Y. Woo measured and analyzed the GIWAXS data. H.S. helped with the ink production. C.M. recorded and analyzed the AFM and c-AFM data. X.L. and M.F. recorded and analyzed the UPS and XPS spectra. Y.P. and W.M.C. performed and analyzed the EPR data. C.-Y.Y. and M.M. fabricated and tested the thermoelectric generator. C.-Y.Y. and D.T. fabricated and tested the OECTs and ternary logic inverters. D.T. designed the ternary inverter and performed SPICE model simulation. C.-Y.Y., M.-A.S., T.-P.R., M.B., and S.F. wrote the manuscript. All authors contributed to discussion and manuscript preparation.

## Funding

## Competing interests

C.-Y.Y., M.-A.S., M.B., and S.F. filed two provisional patent applications related to this work (applications numbers PCT/EP2020/082815 and PCT/EP2020/082821, filed 20 November 2020), and founded n-Ink AB. The other authors declare no competing interests.
