## [Peer Review File · Nature Communications]

Reviewer #1 (Remarks to the Author):

The manuscript by Yang et al. reported a high-conductivity n-type polymeric ink based on BBL. Enough experimental results have assisted their projects. The processing methods are cleverly and the applications in thermoelectric generators and organic electrochemical transistors indeed could pave their realization. The authors have covered all means for the conclusion to give an outstanding analysis of this article. I congratulate the authors for their great effort towards realizing the real thermoelectric generators. Therefore, considering the article's novelty, I recommend this article to be published on Nature Communications. But the authors still have some questions need to be noted.

1. Page 7. The authors claimed that "higher PEIE content leading to a degradation of the electrical performance". But in Figure 3a and Figure S13, the stability of BBL doped with PEIlin and PEIbra is better than PEIE. This should be better illustrated.
2. Page 8. The authors investigated the stability of BBL:PEIlin exposed to common organic solvents in Figure 3c. Although the film thickness has been considered in the calculation of conductivity, we hope the authors could provide the data of film thickness change in Supporting Information to help us understand the process better.
3. Supporting Information, Page 17. In the Supplementary Figure 22, the authors calculated the spin density of BBL doped with PEIlin and PEIbra. But the EPR line shapes of BBL doped with PEIlin and PEIbra are a little different. I thought the line shape of BBL doped with PEIlin is asymmetry. Therefore, the author could give some references under the Supplementary Figure 22.
4. Supporting Information, Page 17. In the Supplementary Figure 23d), the authors gave the UV-Vis absorption spectra of 50wt-% BBL:PEIlin, BBL:PEIbra and BBL:TDAE calculated by subtracting the normalized pristine BBL absorbance. But shouldn't the Y-axis show the numbers because the results only move up. For example, the difference between the intrinsic and 50wt-% BBL:PEIlin at 600-1050 nm only 0.2 from Figure S23a, while from Figure S23d, the difference is about 0.6.
5. The authors should be careful to check their references and correct some abbreviations of the journals.

Reviewer #2 (Remarks to the Author):

The need for an n-type companion to PEDOT-PSS remains unfilled. Its key advantages are stated to be conductivity of 1000 S/cm, processability from benign solutions, and ambient operational stability. The present manuscript reports significant progress toward such an analog based on a novel combination of a previously known polymer (ref 37) and a known polymeric n-dopant (ref 38). The polymeric nature of both components adds to the analogy with PEDOT-PSS. The work deserves prominent mention in the printed/flexible electronics community, at least for the insightful material design. The extent to which performance targets are met is evaluated below.

The ink made from the polymer blend is processed from alcohol solvents. While not quite as benign as aqueous solution, this is a vast improvement in safety and environmental friendliness over halogenated and aromatic solvents generally needed for n-type polymers. The spray-coating process is simple; the most difficult aspects are handling of highly acidic methanesulfonic acid as a processing solvent and a thermal or vacuum anneal (2 hours at 150 degrees C) after coating.

The conductivity of 8 S/cm with thermal, organic solvent, and environmental stability is good for an n-type polymer, though two orders of magnitude less than PEDOT-PSS. The surface being conductive and making ohmic contacts to a variety of electrodes are advantageous.

The ambient temperature conductivity stability (20% decrease in one day in air; stability over 120 days in inert atmosphere) is promising, but that alone may not be a major advance toward ambient atmosphere application compared to other recent literature. For example, conductivity stability of doped n-type polymers with similar order of magnitude conductivity and longer storage times in air has recently been reported: Shi et al. *Adv. Electron. Mater.* 2017, 3, 1700164 and Han et al. *Adv. Funct. Mater.* 2020 <https://doi.org/10.1002/adfm.202005901>. The conductivity stability achieved using the environmentally friendly solvent and process could be considered unique, and should be stated that way, while citing the previous examples of n-type conductivity stability in polymer and flexible materials.

Conductivity stability to high temperature excursions (350 degrees C) and exposure to organic solvents is impressive, but this is in inert atmosphere. If there are applications where packaging and spray processes would keep the atmosphere inert, they should be mentioned.

The authors are to be commended for including the supplementary tables for benchmarking with prior literature. However, the power factor of $11 \mu\text{W m}^{-1} \text{K}^{-2}$ could fairly be considered “moderately high” compared to other examples listed in Table S2, rather than “among the highest reported”. Also, It would be helpful if at least some electrochemical transconductance values were included in Table S5. N-type OECTs have rarely if ever been studied, so careful comparison of transconductance with other OECT examples would be helpful. It is likely that the transconductance is 1-2 orders of magnitude less than for PEDOT-type polymers, but geometry and mobility differences could contribute to this difference.

The various physical characterizations support the main conclusions.

In summary, this is a creative polymer combination to achieve an interesting mix of properties. While some claims could be put into better context, the paper is worthy of publication in *Nature Communications*.

ANSWERS TO THE REVIEWER COMMENTS

Reviewer #1 (Remarks to the Author):

The manuscript by Yang et al. reported a high-conductivity n-type polymeric ink based on BBL. Enough experimental results have assisted their projects. The processing methods are cleverly and the applications in thermoelectric generators and organic electrochemical transistors indeed could pave their realization. The authors have covered all means for the conclusion to give an outstanding analysis of this article. I congratulate the authors for their great effort towards realizing the real thermoelectric generators. Therefore, considering the article's novelty, I recommend this article to be published on Nature Communications. But the authors still have some questions need to be noted.

We appreciate the reviewer's thoughtful comments giving us the possibility to strengthen our manuscript. In the following we address their remarks:

1. Page 7. The authors claimed that "higher PEIE content leading to a degradation of the electrical performance". But in Figure 3a and Figure S13, the stability of BBL doped with PEI_{lin} and PEI_{bra} is better than PEIE. This should be better illustrated.

As correctly pointed out by the reviewer, BBL:PEIE exhibits lower electrical performance, especially at higher PEIE content, compared to BBL:PEI_{lin} and BBL:PEI_{bra}. Unlike PEI, PEIE contains a high concentration of hydroxyl (-OH) groups in its branched structure that could act as trapping sites for electrons (e.g., *Nature* **434**, 194-199 (2005); *Adv. Mater.* **30**, 1800017 (2018)). This could justify the lower electrical conductivity observed at high PEIE content. We discuss this in more details in the caption of Supplementary Figure 13.

2. Page 8. The authors investigated the stability of BBL:PEI_{lin} exposed to common organic solvents in Figure 3c. Although the film thickness has been considered in the calculation of conductivity, we hope the authors could provide the data of film thickness change in Supporting Information to help us understand the process better.

This is an excellent comment. Indeed, the conductivity values reported in Fig. 3c were already normalized for the film thickness. In order to address the reviewer comment, we report in the new Supplementary Table 3 (see Table R1 below) the film thickness changes after washing the layers with the different solvents. Chloroform, chlorobenzene, 1,8-diiodooctane and DMF do not significantly change the BBL:PEI_{lin} film thickness, while strong polar solvents as DMSO and water slightly reduce the film thickness, most likely by removing part of PEI.

Table R1. Thickness of BBL:PEI_{lin} (50% PEI) films before and after solvent washing. Solvent abbreviations: CB = chlorobenzene, DIO = 1,8-diiodooctane, DMF = dimethylformamide, DMSO = dimethylsulfoxide.

Solvents	Thickness before washing (nm)	Thickness after washing (nm)
CHCl ₃	102.2±5.4	107.3±3.5
CB	103.4±2.9	106.5±5.6
DIO	99.1±2.6	102.1±4.7
DMF	107.3±4.3	104.2±2.4
DMSO	100.8±5.8	90.9±4.2
H ₂ O (air)	97.3±4.1	69.1±3.0

3. Supporting Information, Page 17. In the Supplementary Figure 22, the authors calculated the spin density of BBL doped with PEI_{lin} and PEI_{bra}. But the EPR line shapes of BBL doped with PEI_{lin} and PEI_{bra} are a little different. I thought the line shape of BBL doped with PEI_{lin} is asymmetry. Therefore, the author could give some references under the Supplementary Figure 22.

We thank the reviewer for giving us the possibility to clarify this point. The observed asymmetric lineshape of the polaron EPR spectra is caused by g-factor anisotropy (Abragam, A. & Bleaney, B. *Electron paramagnetic resonance of transition ions*, Oxford University Press, Oxford, 1970), which is commonly observed in conductive polymers (*Synth. Met.* **108**, 173 (2000); *J. Phys. Chem. B* **112**, 10922 (2008)). However, we would like to stress that the asymmetric lineshape has already been taken into account in the spin density calculation.

In revision, we added the above discussion to the caption of Supplementary Figure 22.

4. *Supporting Information, Page 17. In the Supplementary Figure 23d), the authors gave the UV-Vis absorption spectra of 50wt-% BBL:PEI_{lin}, BBL:PEI_{bra} and BBL:TDAE calculated by subtracting the normalized pristine BBL absorbance. But shouldn't the Y-axis show the numbers because the results only move up. For example, the difference between the intrinsic and 50wt-% BBL:PEI_{lin} at 600-1050 nm only 0.2 from Figure S23a, while from Figure S23d, the difference is about 0.6.*

This is an excellent comment. Supplementary Figure 23d in the original manuscript was in fact a stacked plot. We agree with the reviewer that this method of plotting the data may be misleading. In revision, we changed this stacked plot with a normal plot to clarify that the absorbance differences between BBL and BBL:PEI_{lin} (or BBL:PEI_{bra}, BBL:TDAE) at 600-1050 nm is about 0.2-0.25. All these three doped samples present very similar absorbance behavior.

5. *The authors should be careful to check their references and correct some abbreviations of the journals.*

In revision, we corrected all references with the right journal abbreviation.

Reviewer #2 (Remarks to the Author):

The need for an n-type companion to PEDOT-PSS remains unfilled. Its key advantages are stated to be conductivity of 1000 S/cm, processability from benign solutions, and ambient operational stability. The present manuscript reports significant progress toward such an analog based on a novel combination of a previously known polymer (ref 37) and a known polymeric n-dopant (ref 38). The polymeric nature of both components adds to the analogy with PEDOT-PSS. The work deserves prominent mention in the printed/flexible electronics community, at least for the insightful material design. The extent to which performance targets are met is evaluated below.

The ink made from the polymer blend is processed from alcohol solvents. While not quite as benign as aqueous solution, this is a vast improvement in safety and environmental friendliness over halogenated and aromatic solvents generally needed for n-type polymers. The spray-coating process is simple; the most difficult aspects are handling of highly acidic methanesulfonic acid as a processing solvent and a thermal or vacuum anneal (2 hours at 150 degrees C) after coating.

The conductivity of 8 S/cm with thermal, organic solvent, and environmental stability is good for an n-type polymer, though two orders of magnitude less than PEDOT-PSS. The surface being conductive and making ohmic contacts to a variety of electrodes are advantageous.

We thank the reviewer for their very positive commentary of our manuscript. In the following we address their remarks. While we agree that the handling of methanesulfonic acid (MSA) as a processing solvent requires extra care, it does not represent a limiting aspect of the ink formulation process. MSA is only used in the initial step of the ink manufacturing to dissolve BBL and it is removed completely during the second step by solvent exchange with ethanol. The use of highly acidic solvent like MSA or sulfuric acid is not new to industry where they are widely employed for the production of Kevlar fibers, for example. Moreover, other solvents like Lewis acids could be used to dissolve BBL and further work in this direction is underway.

Furthermore, the requirement of a thermal activation step under inert atmosphere is not crucial, as this could potentially be performed after encapsulation. It is also noteworthy to mention that the cited thermal annealing conditions (2 hours at 150 degrees C) are required to achieve the highest performances (i.e., conductivity of 8 S/cm). However, for some applications, like for example as an extraction layer in organic and/or hybrid solar cells, a lower conductivity is required. As shown in the Supplementary Figure 12, an annealing temperature as low as 50°C and a thermal annealing time of 1 min is enough for the conductivity to approach 0.1 S/cm.

The ambient temperature conductivity stability (20% decrease in one day in air; stability over 120 days in inert atmosphere) is promising, but that alone may not be a major advance toward ambient atmosphere application compared to other recent literature. For example, conductivity stability of doped n-type polymers with similar order of magnitude conductivity and longer storage times in air has recently been reported: Shi et al. Adv. Electron. Mater. 2017, 3, 1700164 and Han et al. Adv. Funct. Mater. 2020

<https://doi.org/10.1002/adfm.202005901>. The conductivity stability achieved using the environmentally friendly solvent and process could be considered unique, and should be stated that way, while citing the previous examples of n-type conductivity stability in polymer and flexible materials.

This is an excellent comment. As correctly stated by the reviewer, it is the combination of processability from environmentally friendly solvent, high conductivity, remarkable thermal and solvent stability, and good air-stability that make this material unique. To address the reviewer comment, we revised the discussion at p.7 on the air stability of BBL:PEI to include previous examples of stable n-type polymers (*Adv. Funct. Mater.* **30**, 2005901 (2020) is now added as Ref. 42).

Conductivity stability to high temperature excursions (350 degrees C) and exposure to organic solvents is impressive, but this is in inert atmosphere. If there are applications where packaging and spray processes would keep the atmosphere inert, they should be mentioned.

As shown in Supplementary Figure 14, the spray process can be done in ambient atmosphere, and the resulting thin films are stable for several days in air before thermal activation. This simplifies significantly the deposition process as the shelf life of the as-cast films does not require immediate encapsulation. This is a critical aspect of our technology which offers numerous advantages compared to the many n-doped polymer systems reported to date. Regarding encapsulation of the doped films, one could borrow from the organic and hybrid solar cells community where several packaging strategies have been developed (e.g., Su, W. in *Printed Electronics: Materials, Technologies and Applications* (ed Zheng Cui) Ch. 8, 287-315 (Wiley/Higher Education Press, 2016)). In revision, we discussed the encapsulation strategy in the caption of Supplementary Figure 17.

The authors are to be commended for including the supplementary tables for benchmarking with prior literature. However, the power factor of $11 \mu\text{W m}^{-1} \text{K}^{-2}$ could fairly be considered “moderately high” compared to other examples listed in Table S2, rather than “among the highest reported”. Also, it would be helpful if at least some electrochemical transconductance values were included in Table S5. N-type OECTs have rarely if ever been studied, so careful comparison of transconductance with other OECT examples would be helpful. It is likely that the transconductance is 1-2 orders of magnitude less than for PEDOT-type polymers, but geometry and mobility differences could contribute to this difference.

We agree with the reviewer comment and amended the cited statement at p.11. In revision, we also updated Supplementary Table 6 of the revised Supplementary Information to include the transconductance data.

The various physical characterizations support the main conclusions.

In summary, this is a creative polymer combination to achieve an interesting mix of properties. While some claims could be put into better context, the paper is worthy of publication in Nature Communications.

Reviewer #1 (Remarks to the Author):

Yang et al. demonstrated a high-conductivity electron transport polymeric ink, which represents significant progress in polymer electronics. This polymeric ink can be easily prepared and solution-processed. The authors have successfully achieved several applications in n-type thermoelectric generators and organic electrochemical transistors using this polymeric ink. Also, the authors have addressed my previous comments. I strongly suggest the manuscript be suitable for publication in Nature Communications.

Reviewer #2 (Remarks to the Author):

The authors have responded well to the points I raised. The paper is now suitable for publication, with my congratulations!

ANSWERS TO THE REVIEWER COMMENTS

Reviewer #1 (Remarks to the Author):

Yang et al. demonstrated a high-conductivity electron transport polymeric ink, which represents significant progress in polymer electronics. This polymeric ink can be easily prepared and solution-processed. The authors have successfully achieved several applications in n-type thermoelectric generators and organic electrochemical transistors using this polymeric ink. Also, the authors have addressed my previous comments. I strongly suggest the manuscript be suitable for publication in Nature Communications.

We thank the reviewer for their very positive commentary of our manuscript.

Reviewer #2 (Remarks to the Author):

The authors have responded well to the points I raised. The paper is now suitable for publication, with my congratulations!

We thank the reviewer for their recognition and recommendation for our manuscript.